# Roles of Nutrient Limitation on Western Lake Erie CyanoHAB Toxin Production

**DOI:** 10.3390/toxins13010047

**Published:** 2021-01-09

**Authors:** Malcolm A. Barnard, Justin D. Chaffin, Haley E. Plaas, Gregory L. Boyer, Bofan Wei, Steven W. Wilhelm, Karen L. Rossignol, Jeremy S. Braddy, George S. Bullerjahn, Thomas B. Bridgeman, Timothy W. Davis, Jin Wei, Minsheng Bu, Hans W. Paerl

**Affiliations:** 1Institute of Marine Sciences, University of North Carolina at Chapel Hill, Morehead City, NC 28557, USA; hplaas@live.unc.edu (H.E.P.); krossign@email.unc.edu (K.L.R.); jbraddy@email.unc.edu (J.S.B.); 2Stone Laboratory and Ohio Sea Grant, The Ohio State University, Put-In-Bay, OH 43456, USA; chaffin.46@osu.edu; 3Department of Chemistry, State University of New York College of Environmental Science and Forestry Campus, Syracuse, NY 13210, USA; glboyer@esf.edu (G.L.B.); bwei101@syr.edu (B.W.); 4Department of Microbiology, University of Tennessee at Knoxville, Knoxville, TN 37996, USA; wilhelm@utk.edu; 5Department of Biological Sciences, Bowling Green State University, Bowling Green, OH 43403, USA; bullerj@bgsu.edu (G.S.B.); timdavi@bgsu.edu (T.W.D.); 6Lake Erie Center, University of Toledo, Oregon, OH 43616, USA; Thomas.Bridgeman@utoledo.edu; 7Key Laboratory of Integrated Regulation and Resource Development on Shallow Lakes, Ministry of Education, College of Environment, Hohai University, Nanjing 210098, China; weijin@hhu.edu.cn (J.W.); hjxyym@hhu.edu.cn (M.B.)

**Keywords:** cyanotoxins, Maumee Bay, Sandusky Bay, *Microcystis*, *Planktothrix*, microcystin, anatoxin

## Abstract

Cyanobacterial harmful algal bloom (CyanoHAB) proliferation is a global problem impacting ecosystem and human health. Western Lake Erie (WLE) typically endures two highly toxic CyanoHABs during summer: a *Microcystis* spp. bloom in Maumee Bay that extends throughout the western basin, and a *Planktothrix* spp. bloom in Sandusky Bay. Recently, the USA and Canada agreed to a 40% phosphorus (P) load reduction to lessen the severity of the WLE blooms. To investigate phosphorus and nitrogen (N) limitation of biomass and toxin production in WLE CyanoHABs, we conducted in situ nutrient addition and 40% dilution microcosm bioassays in June and August 2019. During the June Sandusky Bay bloom, biomass production as well as hepatotoxic microcystin and neurotoxic anatoxin production were N and P co-limited with microcystin production becoming nutrient deplete under 40% dilution. During August, the Maumee Bay bloom produced microcystin under nutrient repletion with slight induced P limitation under 40% dilution, and the Sandusky Bay bloom produced anatoxin under N limitation in both dilution treatments. The results demonstrate the importance of nutrient limitation effects on microcystin and anatoxin production. To properly combat cyanotoxin and cyanobacterial biomass production in WLE, both N and P reduction efforts should be implemented in its watershed.

## 1. Introduction

Freshwater ecosystems are critical for sustaining life and supporting civilizations throughout history [1]. As the global human population grows, increased urbanization, agricultural and industrial productions, combined with insufficient wastewater treatment practices, have led to a widespread increase in nutrient pollution of these ecosystems, threatening clean and safe water supplies [2]. Excessive inputs of nitrogen (N) and phosphorus (P) have accelerated eutrophication, the process of increasing organic enrichment, which is largely attributable to increased microalgal and aquatic macrophyte growth [3]. The major detrimental impacts of eutrophication include harmful algal bloom (HAB) formation, decreased water transparency (increased turbidity), O_2_ depletion, and reduced biodiversity [3,4]. HAB formation has been a major water quality issue in the U.S. since the 1960s, as noted in a 1965 White House Report indicating HABs as a major source of environmental degradation [5]. Furthermore, nutrient-driven eutrophication of lakes and rivers is one the most significant causes of water quality decline globally [3,6,7,8]. In particular, there are growing concerns about the proliferation and diversification of N- and P-based fertilizers, as they are potent stimulants of aquatic primary production along the freshwater to marine continuum [9,10]. Additionally, climate change (e.g., warming and changing precipitation patterns) is increasing the likelihood of more expansive blooms, exposing human and animal populations (e.g., pets, wildlife, cattle, fish, birds) to water-borne and aerosolized toxins [7,11,12,13,14]. Despite CyanoHAB toxicity being a major human and ecosystem health hazard, the causes and controls of underlying toxicity mechanisms remain poorly understood [15].

Blooms of cyanobacteria in Lake Erie, largely dominated by filamentous heterocystous (N_2_-fixing) forms (*Anabaena/Dolichospermum*, *Aphanizomenon)*, were common in the late 1950s through to the 1970s. These blooms dissipated following the signing in of the *Great Lakes Water Quality Agreement* of 1972, which was updated in 2012. However, the blooms returned as non-N_2_ fixing *Microcystis* blooms in the early 2000s, which have continued and perhaps worsened [16,17], leading to major environmental degradation and increased human health risks [7]. In August 2014, a toxic *Microcystis* spp. bloom in Western Lake Erie (WLE) created a water crisis, forcing public water supplies to be shut down for over 400,000 people in Toledo, OH, USA [7,18]. Nutrient runoff from agricultural nonpoint sources has been a major factor promoting CyanoHABs in WLE [7]. Primary production in Maumee Bay of Lake Erie (largely dominated by *Microcystis* spp. in the summer) shifts from P-limitation to N-limitation with spatial nutrient limitation heterogeneity with N- and P-limitation occurring several km apart [19,20,21]. Prior studies revealed that during the summer months, N was often depleted in embayments such as Sandusky and Maumee Bay [22,23,24,25,26], where summertime molar N:P ratios for Sandusky Bay remained below the canonical Redfield ratio (16:1) [26,27,28]. This suggests the presence of strong N sinks, mediated by denitrification and/or active N cycling and N uptake by high amounts of algal biomass [28,29,30]. The primary fertilizers used in the agriculturally dominated drainage basin of Lake Erie are inorganic fertilizers (ammonium nitrate, urea, and phosphate) and manure, which has low N:P ratios (~5:1), is about 20% [31,32,33,34]. There is an urgent need to determine the linkage between different bioreactive forms of N and P and the promotion of toxic CyanoHABs, to establish the necessary reduction in these nutrient forms to ensure the security of surface potable water. Nutrient reduction will likely need to be even greater as climate change increases the N and P reduction thresholds required for CyanoHAB mitigation [35,36]. The *in situ* bioassay-based study reported here is among the first to use an experimental approach to investigate the response of a natural CyanoHAB community dominated by either *Microcystis* (Maumee Bay) or *Planktothrix* (Sandusky Bay) to actual reductions in N, P or both, under natural conditions in Lake Erie. Satellite and field images of the 2019 WLE blooms can be seen in Figure 1.

A recent review suggested that management efforts to reduce P pollution without controlling N inputs have caused nutrient imbalances in eutrophic systems, which may favor toxic CyanoHABs incapable of fixing atmospheric N_2_ gas, i.e., requiring combined N sources [24]. Prior to P load reductions in the 1970s, CyanoHABs in Lake Erie were mostly the N_2_-fixers *Aphanizomenon* and *Dolichospermum*, formerly *Anabaena* [38]; now, CyanoHABs are primarily non-N_2_-fixing *Microcystis* and *Planktothrix* [16]. In WLE, molecular analysis of the *Microcystis* community indicates a shift from toxic to non-toxic strains that correlates with NO_3_ availability [39], although there appears to be a temporal disconnect as a multiyear analysis found no correlation between the proportion of microcystin-producing genotypes of *Microcystis* and the concentration of microcystin [40]. Recent work has strengthened links between N availability, dominant strain shifts, and toxicity by showing seasonal trends in these patterns [24]. The inability of these cyanobacteria to fix atmospheric N_2_, and their strong affinity for reduced N forms (e.g., NH_4_ and urea), suggests that N delivered through agricultural runoff and internal N recycling mechanisms play critical roles in modulating total phytoplankton biomass, CyanoHAB community composition, and toxicity [39,41].

The prominent cyanotoxins, microcystin and anatoxin, have molecular structures containing N, suggesting that their syntheses may be linked to N availability; hence, there is a need to investigate the potential roles N fertilizers (i.e., NH_4_, NO_3_, and urea) play in bloom dynamics and toxin production in Lake Erie [23,36,42]. A recent study showed that there are N concentration reduction thresholds at which bloom microcystin levels will decrease, leading to further evidence that N limitation may play a role in controlling cyanotoxin production in the WLE blooms [41]. Due to the shift to non-N_2_-fixing CyanoHABs, a major unknown concerning this shift in nutrient limitation is how specific microcystin and anatoxin production potentials are linked to nutrient input reductions.

The US Environmental Protection Agency (EPA) and Environment and Climate Change Canada have recommended a 40% reduction in springtime P loading into WLE to help control the blooms [43,44,45,46]. The 40% P load reduction was the result of a multiple modeling exercise included in the Great Lakes Water Quality Agreement between the US and Canada [47]. As both N and P have been shown to influence the WLE CyanoHABs, it is crucial to investigate the effects of both 40% reductions in both N and P in addition to the investigations of the effects of N and P addition. Here, we addressed the following questions: (1) how do nutrients influence WLE microcystin and anatoxin production? (2) Do the same nutrients limit toxin production and CyanoHAB biomass? (3) Will the 40% P reduction as recommended by the US EPA be effective in reducing CyanoHAB microcystin and anatoxin and biomass production in WLE? (4) Is P reduction alone enough to decrease WLE CyanoHAB biomass and microcystin and anatoxin production, or is a combined N and P reduction strategy needed? Given the relatively high content of N in the cyanotoxins microcystin and anatoxin, we predicted that cyanotoxin production is N-limited and that excessive N inputs promote toxicity of these non-N_2_-fixing CyanoHABs.

## 2. Results

### 2.1. June 2019 Experiement

June 2019 bioassay experiments were characterized by a late spring diatom bloom shortly before the onset of a summer *Microcystis* bloom in Maumee Bay and the very early Planktothrix bloom in Sandusky Bay. In both Maumee and Sandusky Bays, there were high N concentrations—over 200 µmol L^−1^ nitrate plus nitrite in Maumee Bay and over 100 µmol L^−1^ nitrate plus nitrite in Sandusky Bay and relatively low P concentrations of 1–2 µmol L^−1^ dissolved reactive phosphorus (DRP) (Table 1).

In the June Maumee Bay experiment, growth rates significantly differed (*p* < 0.001) among nutrient treatments, but there was no difference between the undiluted and diluted treatments (*p* = 0.76). The +P and +P&N treatments resulted in a higher growth rate than the control and +N treatments, indicating P-limited growth, in both the undiluted and 40% dilution treatments, likely due to the high concentrations of N in the bay (Figure 2; Table 1, Appendix A). Cyanotoxins were not detected in the June Maumee Bay experiment.

In the June Sandusky Bay experiment, nutrient enrichment did not impact growth rates (*p* = 0.68), but growth rates were lower in the 40% diluted treatments (*p* = 0.013); Figure 2; Appendix A), which indicates nutrient-replete conditions. The initial undiluted total microcystin concentration was 0.136 µg/L and total anatoxin concentration was 0.053 µg/L (Appendix A). Microcystin concentrations increased throughout the experiment in the no dilution treatments but not the 40% dilution (Figure 3). The Sandusky Bay microcystin production rate was slightly yet not significantly affected by nutrient enrichment (*p* = 0.067), becoming significant in the biomass-normalized analyses (*p* < 0.001). However, the production rate was lower in the 40% diluted treatments (Figure 3 and Appendix A). The June 2019 Sandusky Bay anatoxin production rate was not affected by dilution treatment with a slight nutrient effect in the biomass-normalized analyses (*p* < 0.01) (Figure 4).

### 2.2. August 2019 Experiment

August experiments were characterized by dense blooms of *Microcystis* in Maumee Bay and *Planktothrix* in Sandusky Bay. Maumee Bay had high N concentrations—over 100 µmol L^−1^ nitrate plus nitrite—but Sandusky Bay had low N concentrations with 6.5 µmol L^−1^ nitrate plus nitrite (Table 2). Both Maumee and Sandusky Bay had low P concentrations of 0.03 to 0.20 µmol L^−1^ DRP (Table 2).

Chlorophyll *a* concentrations decreased throughout incubation of the very dense bloom in the August Maumee Bay experiment, coinciding with negative growth rates (Figure 5). The diluted treatments had reduced algal mortality compared to the undiluted treatments likely due to lower initial starting biomass (*p* < 0.001). The growth rate was significantly affected by nutrients (*p* < 0.001) and the interaction between nutrients and dilution (*p* = 0.012), but there was no discernable pattern, leading to a lack of ecological significance. The initial undiluted total microcystin concentration in the August Maumee Bay experiment was 18.06 µg/L. Microcystin concentration and production rates (Figure 6) followed a similar pattern to chlorophyll with a non-significant nutrient effect (*p* = 0.14) and significant dilution effect without biomass-normalization (*p* < 0.001) and a non-significant effect in biomass-normalized analysis (*p* = 0.5452). Anatoxin was not detected in the Maumee Bay August experiment.

In the August Sandusky Bay experiment, chlorophyll concentration increased throughout the incubation in the three N-only treatments and the + N&P treatment, while it declined in the control and P-only treatment in both the diluted and non-diluted treatments (Figure 5), which indicates N was the primary limiting nutrient. The various forms of N did not exert a discernable difference on growth rates. The highest growth rates were measured in the +N&P treatments, which indicates a secondary P limitation. The dilution effect was also significant (*p* = 0.004). The initial undiluted anatoxin concentration was 0.596 µg/L (Figure 7). Anatoxin production was primarily N-limited both with and without biomass normalization (*p* < 0.001), like growth rates, but P was not secondarily limiting. Unlike chlorophyll, which decreased throughout incubation in the control and P-only treatment, anatoxin concentrations in the control and P-only treatment remained constant throughout the incubation due to production rates of anatoxin increasing throughout the incubation.

## 3. Discussion

Given that CyanoHABs and their associated cyanotoxins have led to adverse human and ecosystem health outcomes in WLE [18], it is important to clarify the major driver(s) of CyanoHAB toxicity. This study investigated nutrient limitation on biomass production and cyanotoxin production, focusing on microcystin and anatoxin. We found that high concentrations of both major nutrients, P and N, drove CyanoHAB growth and microcystin and anatoxin production in WLE. We also found times when the 40% reduction in nutrients could slow microcystin production during nutrient replete conditions (Figure 3E,F).

We found that the June 2019 late spring diatom bloom in Maumee Bay was P-limited, which was induced in both the undiluted and 40% dilution samples due to high ambient N concentrations (>100 µmol/L), while the June 2019 Sandusky Bay *Planktothrix* bloom was not affected by nutrient addition, but growth was slowed following a 40% reduction in nutrients. This is possibly explained by the rapid growth associated with the early bloom, with the 40% reduction in nutrients dropping below the threshold needed to support this bloom [48]. During the bloom maxima in August 2019, the Maumee Bay *Microcystis* bloom was nutrient replete under both undiluted and 40% dilution treatments, with less of a decline in the biomass due to the 40% lower starting biomass following dilution. Additionally, ammonium concentration was higher in the initial 40% dilution than the undiluted sample in both the June and August 2019 Maumee Bay, likely due to an initial die off in the subsample, leading to increased regenerated N as ammonium. These results are likely due to bottle effects attributable to the very high biomass; restricted exchange of gases and nutrients [49,50,51]. The August Sandusky Bay *Planktothrix* bloom was N-limited in both the 40% reduction and the undiluted samples. All nutrient concentrations in the August 2019 Sandusky Bay 40% dilution were higher than concentrations in the undiluted treatment, likely due to the rapid growth of the *Planktothrix* bloom using up more nutrients in the undiluted control group prior to sample filtration, when compared to the reduced biomass in the 40% dilution. Differences between the effects of the different N species were not significant at either location during either experimental period, which has been seen previously in strongly N-limited blooms in WLE [52], but differs from past findings in WLE during periods of weaker N-limitation [22,53,54,55]. This could be due to the high ambient concentrations of NO_3_ paired with low NH_4_ (Table 1 and Table 2). Our findings of N limitation contradict the previous assumption that P availability exclusively controls CyanoHABs [56,57,58,59]. Instead, these findings support the paradigm shift to also consider N input reductions to mitigate CyanoHABs [19,29,60,61].

During the early Sandusky Bay *Planktothrix* bloom (June 2019), microcystin production shifted from between N and P co-limitation in the undiluted samples to nutrient deplete conditions in the 40% dilution samples. This is likely due to the bloom’s use of nutrient resources early on to support biomass production rather than produce secondary metabolites, e.g., cyanotoxins, possibly due to the genetic inability of the June populations to produce the microcystin as seen in prior years [62,63]. Alternatively, the cells could have lysed due to viral or other processes and the dissolved microcystin was not captured on the 0.7 µm porosity GF/F filters or degraded [64,65]. At its peak in August 2019, the *Microcystis* bloom in Maumee Bay was the only bloom that produced microcystin. This production of microcystin occurred under nutrient replete conditions, with less of a decline in microcystin concentrations with slight P limitation in the diluted samples and no apparent nutrient limitation in the undiluted samples. Neither experiments showed significant effects of the various forms of N.

Even though cyanobacteria require N to produce N-rich microcystin, P is also required for cellular growth to allow for higher microcystin concentrations. As the ratio of microcystin to chlorophyll *a* in both June and August was nearly linear (Figure 3 and Figure 6), we conclude that the primary bloomers—*Planktothrix* in Sandusky Bay and *Microcystis* in Maumee Bay—were the primary producers of microcystin. The P requirement for microcystin production has been observed in prior studies in Lake Erie, and in several German lakes [66]. This deviates from previous studies that clearly demonstrated links between N availability and higher N:P and bloom toxicity in microcystin-producing blooms [7,67,68,69]. This could be due to microcystin being an “N bargain” with a C:N ratio of 4.9:1 compared to the average of 3.6:1 in a survey of 2000 proteins [69]. However, P-limitation of microcystin production has been shown to occur in chemostat experiments [70] and in a transcriptome experiment on Lake Erie blooms [40]. The microcystin congener pattern observed in these experiments followed what was expected for North American lakes, including Lake Erie, with microcystin LR, YR, RR being the dominant congeners [18].

We observed anatoxin production in the Sandusky Bay *Planktothrix* bloom during both early and peak blooms. This is the first study showing anatoxin production in Lake Erie, although it has been shown that anatoxin production can occur during *Planktothrix* blooms accompanied by other cyanobacteria, including *Cuspidothrix issatschenkoi*, which has previously been identified in Sandusky Bay [23,71,72,73,74,75]. This was likely the case, as the biomass normalized anatoxin production mirrors the anatoxin production in the non-normalized analysis (Figure 4 and Figure 7), meaning that secondary cyanobacterial species may be driving the anatoxin production in Sandusky Bay. During the early *Planktothrix* bloom in June 2019, there was no apparent nutrient limitation in the undiluted treatments. However, there was co-limitation by both N and P in the diluted treatments. During the peak bloom in August 2019, anatoxin production was N-limited in both the undiluted and 40% diluted samples. While no differences were found between forms of N added in the June bioassay, during the peak bloom in August, NO_3_ additions led to higher concentrations of anatoxin compared to NH_4_ and urea additions. Additionally, N limitation of anatoxin production has been shown previously [76]. As observed in this experiment, higher overall N concentrations lead to higher anatoxin concentrations, with NO_3_ enrichment leading to the largest increase in anatoxin production, which parallels results from other limnetic anatoxin-producing CyanoHABs [77,78,79,80,81]. Anatoxin production in Sandusky Bay and other *Planktothrix*-dominated bodies of water needs further examination, given the neurotoxicity and potential developmental toxicity of anatoxin [82,83] as well as its multiple deleterious environmental effects [84,85].

Nutrient concentrations were very high during both the early and peak 2019 bloom in Maumee Bay with 223.67 ± 25.43 µg L^−1^ NO_3_ and 2.224 ± 1.008 µg L^−1^ DRP in June and 127.12 ± 10.82 µg L^−1^ combined NO_3_ and NO_2_ in August and 0.203 ± 0.199 µg L^−1^ DRP. Similar to Maumee Bay, Sandusky Bay exhibited high nutrient concentrations in June with 101.45 ± 5.95 µg L^−1^ NO_3_ and 0.203 ± 0.138 µg L^−1^ DRP in June, but had lower nutrient concentrations in August with 127.12 ± 10.82 µg L^−1^ NO_3_ in August and 0.032 ± 0.012 µg L^−1^ DRP. This is likely due to larger nutrient loads from the Maumee River than from the Sandusky River, as seen previously in 2007 [86]. The high nutrient loads were exacerbated by elevated precipitation associated with a very wet winter in 2019 [87], which will likely continue to be an issue as high precipitation events are predicted to continue in the future [88,89,90]. Denitrification and assimilation draw down nitrate to concentrations below the threshold of detection (<0.5 µmol/L) throughout summer and fall in western Lake Erie and Sandusky Bay [28,91], which is a pattern that occurs independent of tributary nutrient loads [19]. Our Maumee Bay experiments occurred before nitrate depletion, and therefore, we would expect to have observed N-limited growth and microcystin production following the N depletion [25]. However, it remains to be seen how a 40% dilution in nutrients (N and P) would affect N-limited *Microcystis* in late summer. Therefore, nutrient input reductions need to target both N and P rather than just P as recommended by the US EPA and Environment and Climate Change Canada [43,44,45,92]. While P reduction is actively pursued [93], N management strategies are required as well [35,94,95].

## 4. Conclusions

Our results suggest that nutrient dynamics play a crucial role in the WLE CyanoHABs for both biomass production as well as microcystin and anatoxin production in the eutrophic Sandusky and Maumee Bays. During the peak bloom periods when microcystin and anatoxin concentrations are highest, microcystin production was nutrient deplete and anatoxin production was N-limited. Maumee Bay biomass shifted from P-limited immediately prior to the *Microcystis* bloom to nutrient deplete during peak bloom, while the Sandusky Bay *Planktothrix* bloom shifted from nutrient deplete to N-limited from early bloom to peak bloom. A 40% reduction in N and P led to a slight reduction in biomass and microcystin and anatoxin production. However, further studies are needed to investigate the long-term nutrient reduction thresholds needed to control CyanoHABs. With N and P enrichment stimulating the WLE CyanoHABs, there is a need to constrain external loads of both N and P, and impose stricter nutrient-limited conditions in order to help mitigate the CyanoHAB problem in WLE [52,96,97,98]. Our study took place only in eutrophic bays and we showed that a 40% reduction might not be enough in Maumee and Sandusky Bay because growth and toxin production could still be nutrient-saturated. Future studies are needed to determine if a 40% reduction is adequate for the open waters of WLE. Furthermore, an adaptive management approach is needed to determine if the 40% reduction goal needs to be adjusted with changes in land use practices and climate change [99]. Additionally, future studies should focus on drawing direct functional links between nutrient enrichment and cyanotoxin production, e.g., Krausfeldt et al. [36]. Lastly, anatoxin should be more closely monitored in WLE, as it is a potent neurotoxin with human health-associated implications [100].

## 5. Materials and Methods

### 5.1. Bioassay Methods

We performed experimental manipulations of natural Maumee Bay (Oregon, OH, USA) and Sandusky Bay (Sandusky, OH, USA) phytoplankton communities that were collected from nearshore docks (Figure 8; Appendix A). Water was pumped from 1 m below the surface into pre-cleaned (flushed with lake water) 20 L carboys using a non-destructive diaphragm pump and was transported to The Ohio State University Stone Laboratory on South Bass Island (Put-in-Bay, OH, USA) (Figure 8).

This experiment deployed in situ bioassays, using 4 L pre-cleaned polyethylene Cubitainers to which natural lake water was added from Maumee and Sandusky Bays using the methodology described in Paerl et al. [101] and Xu et al. [102]. Microcosm treatments were individually amended with either 100 μM N of NO_3_ (as KNO_3_), 100 µM N of NH_4_ (as NH_4_Cl), 6 µM PO_4_ (as KH_2_PO_4_), 100 µM N and 6 µMP added as a combined addition of 50 µM NO_3_, 50 µM NH_4_, and 6 µM PO_4_, and, in August 2019, urea (50 µM urea to achieve 100 µM N), yielding similar total dissolved nutrient concentrations (for each treatment) and falling within a range matching riverine dissolved inorganic nutrient discharge into Lake Erie nearshore waters. To avoid silica or dissolved inorganic carbon limitation in Cubitainers during the incubation period, we added 50 µM Si as Na_2_SiO_3_ and 10 mg L^−1^ (83.25 µM) DIC as NaHCO_3_ based on previous Si and DIC values from Hanson et al. [103] and Rockwell et al. [104]. We used a major ion solution (MIS) specific to WLE to provide 40% dilutions to mimic the EPA-recommended reductions in P inputs to WLE as well as a parallel 40% reduction in N, as both N and P have been shown to influence WLE CyanoHAB bloom dynamics [22,39,43]. The 40% dilution control investigated a 40% reduction in both N and P. Incubations were run for 72 h at a lake site near the Stone Laboratory at ambient lake water temperatures and light conditions [23,101,102]. Based on previous work on eutrophic Lake Taihu, China [95], a 72 h maximum incubation period was chosen to minimize “bottle effects”, while having ample time to examine phytoplankton growth, microcystin, and anatoxin production responses.

To perform nutrient dilutions, we developed a major ion solution (MIS) for WLE, which provided a N- and P-free dilution media to minimize hypertonic and hypotonic effects on the organisms in the samples by balancing major dissolved ions in the system (Table 3). As an example, artificial seawater is the MIS for the open ocean. For WLE, we based the ambient ion concentrations on a past study by Chapra et al. [105]. As there is substantial natural variability due to rainfall and evaporative effects and the ions in the MIS are in micromolar concentrations and pulse events change the ions in WLE, these deviations are considered reasonable. The compounds used in the MIS are found in Appendix A.

### 5.2. Phytoplankton Biomass Determination

Chlorophyll *a*, as an indicator of phytoplankton biomass, was measured on subsampled samples by filtering 50 mL of sample water onto Whatman glass fiber filters (GF/F). Filters were frozen at −20 °C and subsequently extracted using a tissue grinder in 90% acetone [107,108]. Chlorophyll *a* in extracts was measured using the non-acidification method of Welschmeyer [109] on a Turner Designs Trilogy fluorometer calibrated with pure Chlorophyll *a* standards (Turner Designs, Sunnyvale, CA, USA).

### 5.3. Nutrient Concentration Determination

Nutrient samples were collected in 50 mL Falcon tubes by collecting the GF/F filtered water from the chlorophyll *a* sample collection and frozen at −20 °C. A continuous segmented flow auto-analyzer (QuAAtro SEAL Analytical Inc., Mequon, WI, USA) was used to quantify nitrate, nitrite, ammonium, dissolved reactive P, and silicate using standard U.S. EPA methods [110]. Urea concentration (as urea-N) was determined spectrophotometrically [52,111,112].

### 5.4. Anatoxin and Microcystin Determinations

Cyanotoxins were measured on subsampled samples by filtering 50 mL of the sample water onto Whatman GF/F. Filters were frozen at −20 °C until extraction with ultrasonic sonication in 5 mL of 50% methanol and 1% acetic acid. Samples were centrifuged at 14,000× *g* for 10 min at 4 °C. The supernatants were filtered through 0.45 µm pore-size nylon syringe filters (Corning, CLS431225) and stored at −20 °C until analysis. Microcystin was quantified via coupled liquid chromatography/mass spectrometry using methods modified from Boyer [113] and Peng et al. [114]. Reverse-phase liquid chromatography using a Waters 2695 solvent delivery system (Waters, Milford, MA, USA) coupled to a Waters ZQ4000 mass spectrometer (Waters, Milford, MA, USA) (m/z 500–1250 amu) and a 2996 photodiode array detector (Waters, Milford, MA, USA) (210 to 400 nm wavelength) was used to screen for molecular ions of 22 common microcystin congeners (RR, dRR, mRR, H4YR, hYR, YR, LR, mLR, zLR. dLR, meLR, AR, FR, WR, LA, dLA, mLA, LL, LY, LW, LF, WR). Separation conditions used an ACE 5 C18, 150 × 3.0 mm column and a 30–70% aqueous acetonitrile gradient containing 0.1% formic acid at a flow rate of 0.3 mL min. Individual congener concentrations were quantified using the peak area of the extracted ion relative to standards of microcystin-LR (Enzo Life Sciences, Ann Arbor, MI, USA). This allows quantification of congeners where standards are not available. Detection of congeners was validated by co-occurring presence of the diagnostic UV signature from the ADDA group. Full methodological details and the standard operating protocols are available from Protocols.io [115].

Anatoxin-a, dihydro-anatoxin-a and homoanatoxin-a were determined by LC-MS/MS using one quantification ion and two confirmation ions for each compound. Separation was achieved with an ACE 5 4.6 × 150 mm column (MacMod Analytical, Chadds Ford, PA, USA) assembly with solvent flow of 0.5 mL/min from a Waters Alliance 2695 solvent system (Waters, Milford, MA, USA). The solvent system was: A, 0.1% formic acid in water; B, 0.1% formic acid in acetonitrile. The separation gradient was: 0 to 20% B from 0 to 10 min, 20% to 80% B from 10 to 20 min, and 80% to 100% B from 20 to 23 min, followed by equilibration back to 0% B for 7 min. Toxins were identified using a Waters Acquity TQD mass spectrometer (Waters, Milford, MA, USA) operated in positive mode with capillary voltage 3.5 kV, desolvation and cone gasses at 30 and 800 Lh^−1^, respectively, desolvation and source temperatures of 400 and 150 °C, respectively. Retention times and fragmentation patterns were determined using anatoxin-a (BioMOL International, Farmingdale, NY, USA), homoanatoxin-a isolated from natural sources and α and β dihydroanatoxin synthesized by catalytic hydrogenation/reduction of anatoxin-a [116]. Calibration was performed with anatoxin-a; dihydro-anatoxin-a and homoanatoxin-a concentrations were estimated using the anatoxin-a standard curve. A phenylalanine standard was run with each set to confirm the baseline resolution between anatoxin-a and phenylalanine. Multiple reaction monitoring quantitation transitions were: anatoxin-a (166.09 > 131.00, collision energy (CE) 15 eV), dihydro-anatoxin-a (168.20 > 43.10, CE 23 eV), homoanatoxin-a (180.10 > 163.10, CE 15 eV). Confirmation transitions were: anatoxin-a (166.09 > 148.90, CE 15 eV; 166.09 > 90.90, CE 17 eV), dihydro-anatoxin-a (168.20 > 55.90, CE 22 eV; 168.20 > 67.00, CE 26 eV), homoanatoxin-a (180.10 > 145.10, CE 15 eV; 180.10 > 105.00, CE 17 eV).

### 5.5. Data Transformation and Analysis

To remove biomass effects on toxin to better measure nutrient effects on microcystin and anatoxin production, microcystin and anatoxin concentrations are normalized to biomass as proxied by chlorophyll *a*. Microcystin:chl *a* and anatoxin:chl *a* ratios are calculated using Equation (1):(1)$$toxin:chl\;a\;ratio\;\left(\mu g\;microcystin\;or\;anatoxin\;\mu g\;chlorophyll\;a^{-1}\right)=\;\frac{\left[toxin\right]}{\left[biomass\right]}$$
where [toxin] is the concentration of either microcystin or anatoxin (in µg L^−1^) and [biomass] is the concentration of chlorophyll *a* (in µg L^−1^).

For comparison between dilution treatments, we calculated production rates from the chlorophyll *a* and biomass-normalized microcystin and biomass-normalized anatoxin concentrations. Production rate (d^−1^) is a method to ln normalize the changes in concentrations, where a production of 0.693 d^−1^ is a doubling of the concentration per day, a production of 0.0 d^−1^ indicates no change, and a production of −0.693 d^−1^ represents a halving of the concentration. Production is calculated using Equation (2):(2)$$Production\;\left(d^{-1}\right)=\;\ln\left(\frac{\mu_{T3}}{\mu_{T0}}\right)*\frac{1}{t}$$
where *µ*_*T*0_ is the average value of the measurement for the initial time point (T0), *µ*_*T*3_ is the average value of the measurement for the time point of 3 days (T3), and t is the time difference between the samplings (in days), which in this case is *t* = 3 days. To calculate the standard deviation for the production, propagated standard deviation is used, as calculated by Equation (3):(3)$$Propagated\;Standard\;Deviation=\;\sqrt{{\left(\frac{\sigma_{T0}}{\mu_{T0}}\right)}^{2}+{\left(\frac{\sigma_{T3}}{\mu_{T3}}\right)}^{2}}$$
where *µ*_*T*0_ is the average value of the measurement for T0, *σ*_*T*0_ is the standard deviation for the measurement at T0, *µ*_*T*3_ is the average value of the measurement for T3, and *σ*_*T*3_ is the standard deviation for the measurement at T3. For error bars, standard error is used, which is calculated using Equation (4):(4)$$Standard\;Error=\;\frac{\sigma }{\sqrt{n}}$$
where *σ* is the standard deviation for Figure 2a–d, Figure 3a–d, Figure 4a–d, Figure 5a–d, Figure 6a–d, and Figure 7a–d, *σ* is the propagated standard deviation for Figure 2e–f, Figure 3e–f, Figure 4e–f, Figure 5e–f, Figure 6e–f, and Figure 7e–f, and *n* is the number of data points. The standard errors are available in the WLE_Barnard_et_al_Toxins GitHub repository [117].

### 5.6. Statistical Analysis

To evaluate the source of the variation between the treatments, ANOVA analyses were performed. For this experiment, two-factor ANOVA analyses were run on balanced data sets (all data *n* = 3), and n-factor ANOVA analyses were run on unbalanced data sets (one or more treatments were characterized as *n* = 1 or *n* = 2) using MATLAB ver. R2018b [118]. Both the two-factor and n-factor ANOVA analyses calculate degrees of freedom (d.f.) as the number of treatments (*n*) minus one (d.f. = *n*−1). The homogeneity of variances was tested for with Levene’s Absolute test using MATLAB ver. R2018b [118]. All data and corresponding n-values are in Appendix A

## Figures and Tables

**Figure 1 toxins-13-00047-f001:**
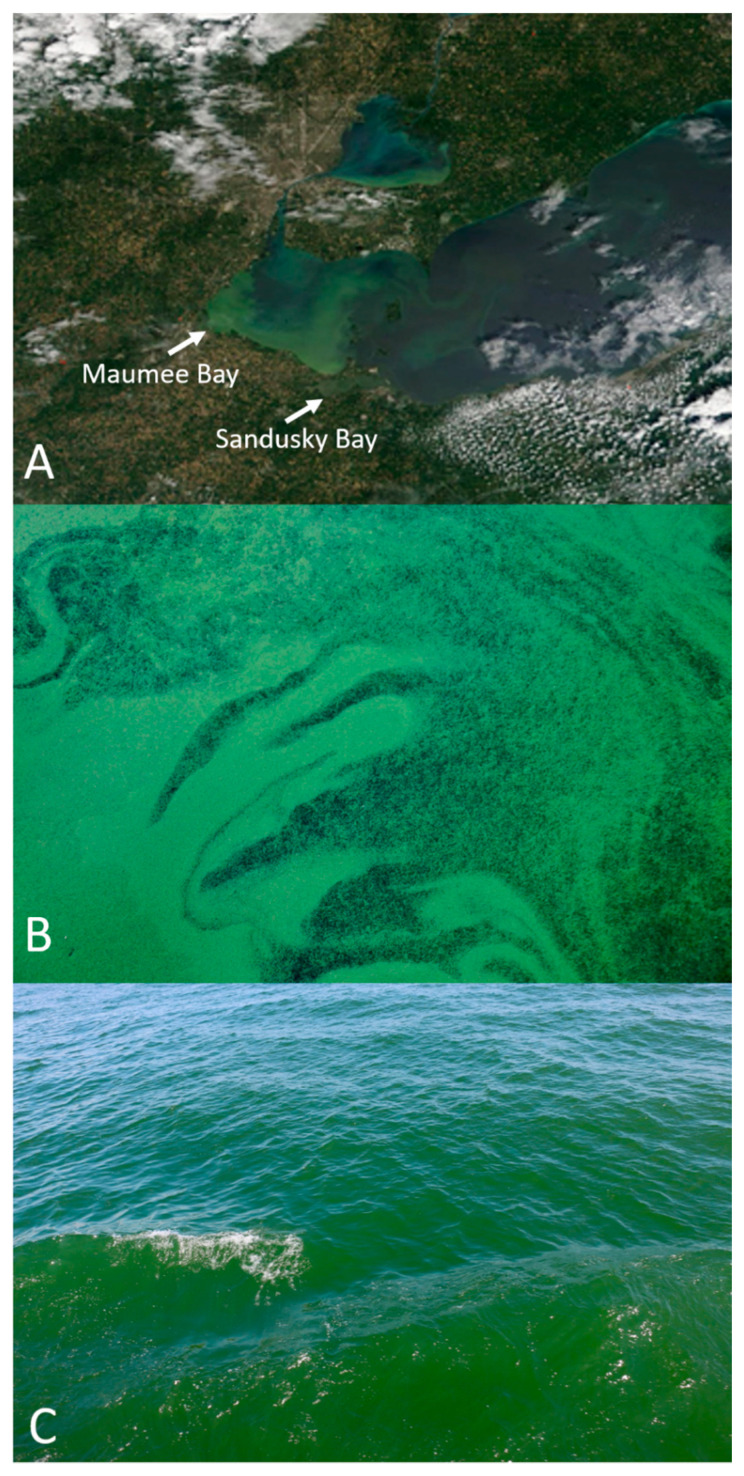
Images of the 2019 WLE CyanoHABs. (**A**) Satellite imagery from the NASA Terra satellite of the WLE CyanoHAB on 19 August 2019 as provided by NOAA MODIS [37]; (**B**) Maumee Bay *Microcystis*-dominated cyanobacterial harmful algal bloom (CyanoHAB) on 4 August 2019 during sampling for the August 2019 bioassays. Photo by H. Plaas; (**C**) Sandusky Bay *Planktothrix*-dominated bloom on 4 August 2019 during sampling for the August 2019 bioassays. Photo by H. Plaas.

**Figure 2 toxins-13-00047-f002:**
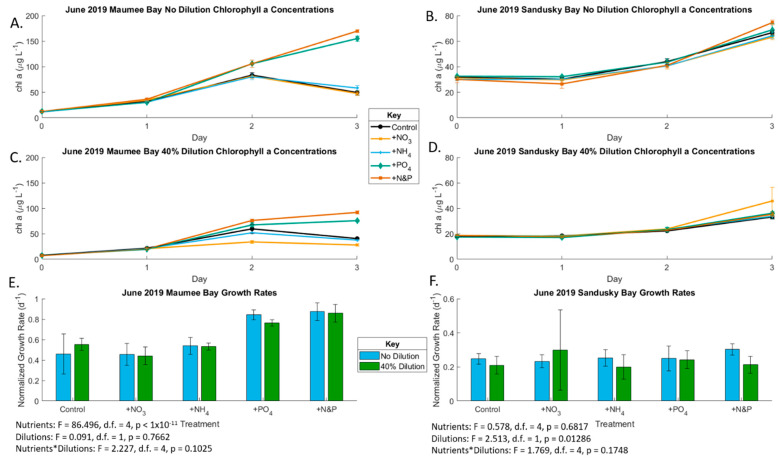
Growth rates of phytoplankton, as determined by chlorophyll *a* accumulation during the course of incubation in the June 2019 bioassays. (**A**) Undiluted Maumee Bay water (also see Appendix A); (**B**) undiluted Sandusky Bay water (also see Appendix A); (**C**) 40% dilution Maumee Bay water (also see Appendix A); (**D**) 40% dilution Sandusky Bay water (also see Appendix A); (**E**) Maumee Bay growth rates under the various nutrient addition treatments at the two locations of T3 compared to T0 (also see Appendix A). Error bars are standard error; (**F**) Maumee Bay growth rates under the various nutrient addition treatments at the two locations of T3 compared to T0 (also see Appendix A). Error bars are standard error. Significances between treatments for (**E**,**F**) are from two-factor ANOVAs.

**Figure 3 toxins-13-00047-f003:**
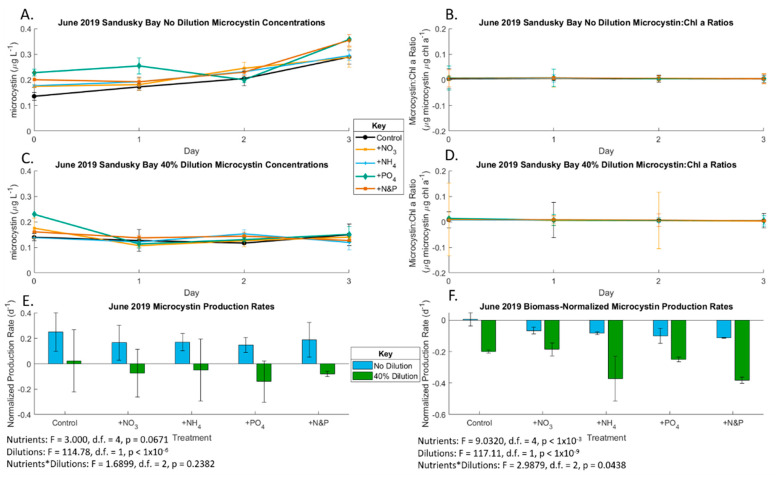
Production rates of microcystin during the June 2019 bioassays. Only Sandusky Bay produced microcystin in June. (**A**) Undiluted Sandusky Bay microcystin concentrations (also see Appendix A); (**B**) undiluted Sandusky Bay biomass-normalized microcystin concentrations (also see Appendix A); (**C**) 40% dilution Sandusky Bay microcystin concentrations (also see Appendix A); (**D**) 40% dilution Sandusky Bay biomass-normalized microcystin concentrations (also see Appendix A); (**E**) Maumee Bay microcystin production rates under the various nutrient addition treatments at the two locations of T3 compared to T0 (also see Appendix A); (**F**) Maumee Bay biomass-normalized microcystin production rates under the various nutrient addition treatments at the two locations of T3 compared to T0 (also see Appendix A). Error bars are standard error. Significance for (**E**,**F**) is from *n*-factor ANOVA analysis due to unbalanced data sets.

**Figure 4 toxins-13-00047-f004:**
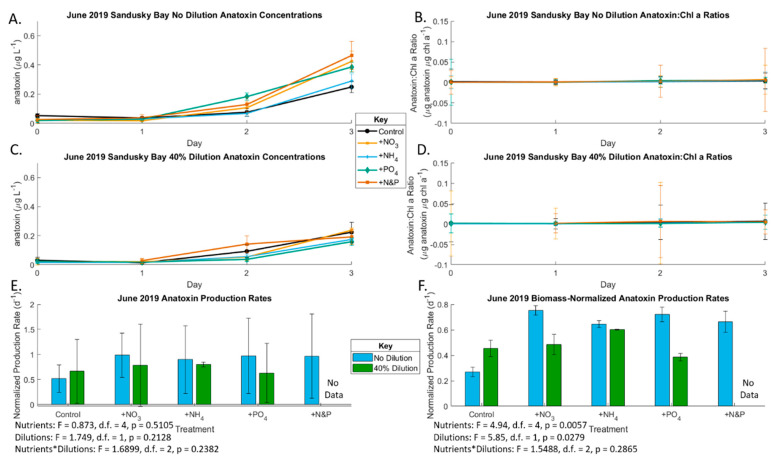
Chlorophyll *a*-based production rates of anatoxin during the June 2019 bioassays. Only Sandusky Bay produced anatoxin in June. (**A**) Undiluted Sandusky Bay anatoxin concentrations (also see Appendix A); (**B**) undiluted Sandusky Bay biomass-normalized anatoxin concentrations (also see Appendix A); (**C**) 40% dilution Sandusky Bay anatoxin concentrations (also see Appendix A); (**D**) 40% dilution Sandusky Bay biomass-normalized anatoxin concentrations (also see Appendix A); (**E**) Maumee Bay anatoxin production rates under the various nutrient addition treatments at the two locations of T3 compared to T0 (also see Appendix A); (**F**) Maumee Bay biomass-normalized anatoxin production rates under the various nutrient addition treatments at the two locations of T3 compared to T0 (also see Appendix A). Error bars are standard error. Significance for (**E**,**F**) is from *n*-factor ANOVA analysis due to unbalanced data sets.

**Figure 5 toxins-13-00047-f005:**
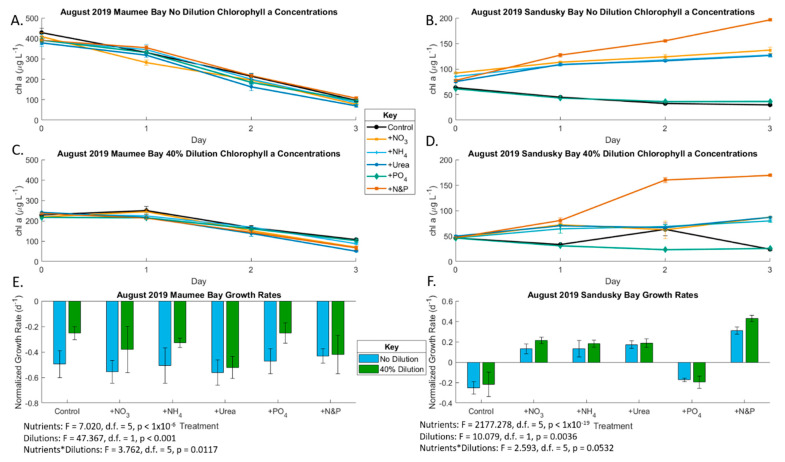
Growth rates of phytoplankton in the August 2019 bioassays. (**A**) Undiluted Maumee Bay Chlorophyll *a* (also see Appendix A); (**B**) undiluted Sandusky Bay Chlorophyll *a* (also see Appendix A); (**C**) undiluted Maumee Bay Chlorophyll *a* (also see Appendix A); (**D**) 40% dilution Sandusky Bay Chlorophyll *a* (also see Appendix A); (**E**) Maumee Bay growth rates under the various nutrient addition treatments at the two locations at T3 compared to T0 (also see Appendix A). Error bars are standard error. Significance for (**E**) is from two-factor ANOVA analysis; (**F**) Maumee Bay growth rates under the various nutrient addition treatments at the two locations at T3 compared to T0 (Appendix A). Error bars are standard error. Significance for (**F**) is from *n*-factor ANOVA analysis due to unbalanced data sets.

**Figure 6 toxins-13-00047-f006:**
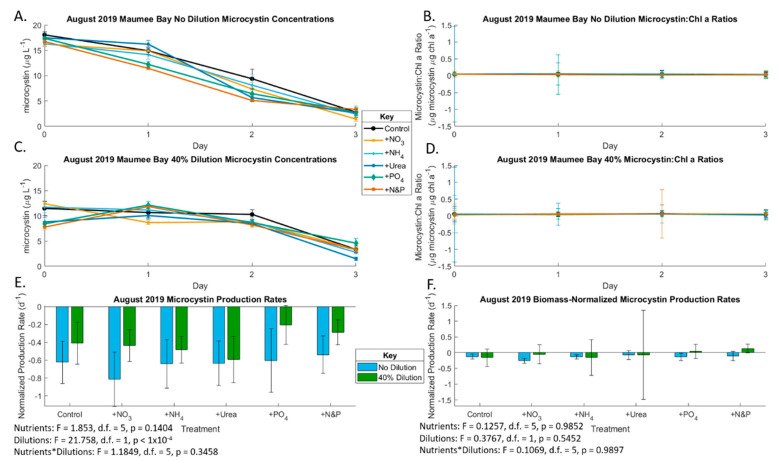
Production rates of microcystin during the August 2019 bioassays. Only Maumee Bay produced microcystin in all samples. (**A**) Undiluted Maumee Bay microcystin concentrations (also see Appendix A); (**B**) undiluted Maumee Bay biomass-normalized microcystin concentrations (also see Appendix A); (**C**) 40% dilution Maumee Bay microcystin concentrations (also see Appendix A); (**D**) 40% dilution Maumee Bay biomass-normalized microcystin concentrations (also see Appendix A); (**E**) Maumee Bay microcystin production rates under the various nutrient addition treatments at the two locations of T3 compared to T0 (also see Appendix A); (**F**) Maumee Bay biomass-normalized microcystin production rates under the various nutrient addition treatments at the two locations of T3 compared to T0 (also see Appendix A). Error bars are standard error. Significance for (**E**,**F**) is from 2-factor ANOVA analysis.

**Figure 7 toxins-13-00047-f007:**
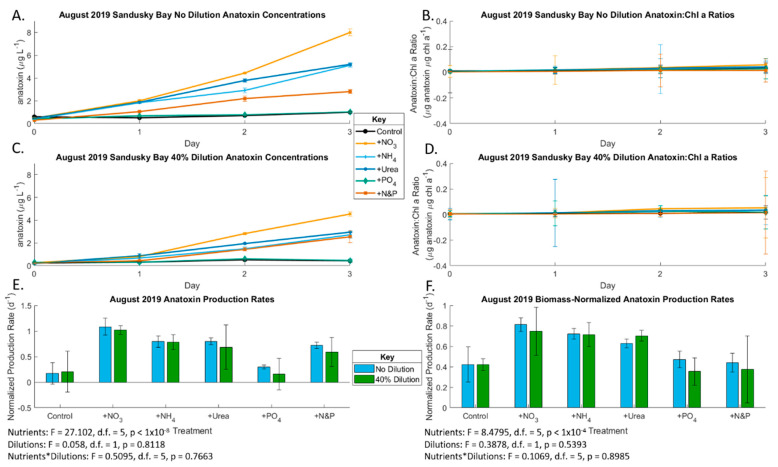
Production rates of anatoxin during the August 2019 bioassays. Only Sandusky Bay produced anatoxin in August. (**A**) Undiluted Sandusky Bay anatoxin concentrations (also see Appendix A); (**B**) undiluted Sandusky Bay biomass-normalized anatoxin concentrations (also see Appendix A); (**C**) 40% dilution Sandusky Bay anatoxin concentrations (also see Appendix A); (**D**) 40% dilution Sandusky Bay biomass-normalized anatoxin concentrations (also see Appendix A); (**E**) Maumee Bay anatoxin production rates under the various nutrient addition treatments at the two locations of T3 compared to T0 (also see Appendix A); (**F**) Maumee Bay biomass-normalized anatoxin production rates under the various nutrient addition treatments at the two locations of T3 compared to T0 (also see Appendix A). Error bars are standard error. Significance for (**E**,**F**) is from 2-factor ANOVA analysis.

**Figure 8 toxins-13-00047-f008:**
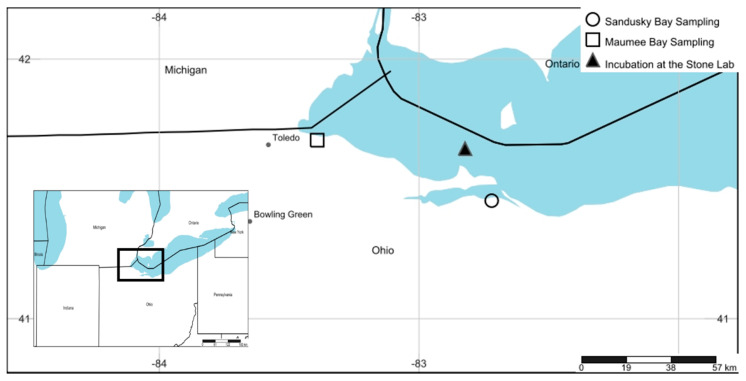
Map of the sampling sites and the location of the incubation. Maumee Bay water was collected off a bulkhead dock near the University of Toledo Lake Erie Center in Oregon, OH, USA. Sandusky Bay sampling took place at a dock outside the Paper District Marina in Sandusky, OH, USA. Incubation took place at The Ohio State Stone Laboratory on South Bass Island (Put-In-Bay, OH, USA). GPS coordinates for the sampling and incubation sites can be found in Appendix A. This figure was created with www.simplemappr.net [106].

**Table 1 toxins-13-00047-t001:** Initial nutrient concentrations in the June 2019 bioassay water collected from control Cubitainers. All data are *n* = 3.

Nutrient Parameter	Maumee Bay		Sandusky Bay	
	No Dilution	40% Dilution	No Dilution	40% Dilution
NO_3_ + NO_2_ (µmol L^−1^)	223.67 ± 25.43	137.64 ± 35.00	101.45 ± 5.95	58.46 ± 8.46
NH_4_ (µmol L^−1^)	1.34 ± 1.01	3.67 ± 0.60	24.28 ± 0.66	17.14 ± 0.85
DRP (µmol L^−1^)	2.24 ± 0.23	1.50 ± 0.07	1.20 ± 0.14	0.85 ± 0.05
Silicate (µmol L^−1^)	139.14 ± 12.23	100.46 ± 2.53	130.42 ± 19.50	78.88 ± 16.20

**Table 2 toxins-13-00047-t002:** Initial concentrations of nutrients in the August 2019 bioassay water taken from T0 control Cubitainers. All data are *n* = 3.

Nutrient Parameter	Maumee Bay		Sandusky Bay	
	No Dilution	40% Dilution	No Dilution	40% Dilution
NO_3_ + NO_2_ (µmol L^−1^)	127.12 ± 10.82	60.10 ± 12.94	6.59 ± 0.29	6.61 ± 0.05
NH_4_ (µmol L^−1^)	0.70 ± 0.42	1.74 ± 1.58	1.05 ± 0.69	1.04 ± 0.06
Urea (µmol L^−1^)	3.45 ± 0.61	1.59 ± 1.15	2.91 ± 1.46	3.99 ± 1.09
DRP (µmol L^−1^)	0.20 ± 0.20	0.05 ± 0.01	0.03 ± 0.01	0.10 ± 0.07
Silicate (µmol L^−1^)	124.33 ± 7.13	95.44 ± 6.24	51.12 ± 12.23	64.91 ± 18.52

**Table 3 toxins-13-00047-t003:** Concentrations of major ions in the ambient Lake Erie water and the major ion solution (MIS) used for the dilutions in the bioassays.

Ion ^1^	Average Ambient Concentration (mg/L) [105]	MIS ^1^ Concentration (mg/L)	MIS ^1^ Concentration (µM)	Percent Difference between Chapra et al. [105] and MIS Concentrations
Ca ^2+^	32.11	32	800	− 0.34%
Mg ^2+^	8.89	8.88	370	− 0.11%
Na ^+^	8.58	4.6	200	− 46.39% ^2^
K ^+^	1.431	1.56	40	9.01% ^3^
Cl ^-^	14.58	16.33	460	12.00% ^3^
SO_4_ ^2-^	22.81	43.2	450	89.39% ^3^

^1^ Constituents of MIS can be found in Appendix A; ^2^ lower concentration compared to ambient concentration; ^3^ higher concentration compared to average ambient concentrations.

## Data Availability

The data formatted for analysis and executable MATLAB code used to produce Figure 2, Figure 3, Figure 4, Figure 5, Figure 6 and Figure 7 can be found on GitHub at www.doi.org/10.5281/zenodo.4281127 [117]. The data presented in this study are also available in table form in the accompanying Appendix A: https://www.mdpi.com/2072-6651/13/1/47/s1.

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
