# Peer review of "Roles of Nutrient Limitation on Western Lake Erie CyanoHAB Toxin Production"

_toxins, 2021, doi:10.3390/toxins13010047_

Round 1
Reviewer 1 Report
The problem of deterioration of drinking water quality is very topical nowadays in many countries of the world. The reason for this is the massive growth of cyanobacteria which produce toxins during their blooming that are dangerous for people and animals. The most dangerous cyanotoxins are hepatotoxic microcystins and neurotoxic paralytic shellfish toxins and anatoxins. In recent years, due to the numerous cases of poisoning of people throughout the world, the task of identification of toxic and toxigenic cyanobacteria became an essential part of monitoring of water bodies. In particular, this applies to economically important water bodies.
Numerous studies were devoted to toxic species, but there is a necessity to study in detail the factors contributing to the massive reproduction of cyanobacteria in water bodies.
The main goal of this study is to elucidate the influence of biogenic factors, such as nitrogen and phosphorus, on the biomass of toxin-producing cyanobacteria and toxin concentration in one of the largest lakes in the world.
The authors make an important conclusion about the reduction in the inflow of both nitrogen and phosphorus to water bodies, which are often mentioned in numerous studies on eutrophication. However, this particular study is distinguished by a thorough analysis and thoughtfulness of the course of experimental work. This solid and elegant experimental work with fundamental conclusions did not cause any remarks from me. In my opinion, it will be useful and interesting for a large number of limnologists and hydrobiologists. It is worth to mention a deep knowledge of the subject by the authors and the neatness of designing the article.
Author Response
Thank you for your comments and we are glad you enjoyed our paper. Please note the following changes in the paper.
Here is a summary of the larger changes made in addition to line edits. We edited the abstract as per reviewer two to better drive home the take away points and key take away points of the manuscript. Additionally, we revised Figures 2-7 to remove the fractional days in the x axes, increase figure panel denotation visibility and increasing the size of the lines and the symbols. We also fixed the number of significant digits in the nutrient tables in addition to adding a discussion of why some nutrient values were higher in the 40% dilution than the no dilution values. Additionally, the conclusion has been rewritten to make the conclusion stronger and to highlight key points.
Reviewer 2 Report
The paper studies the effects of nutrient addition and limitation on algal biomass and cyanotoxin production, focusing on microcystin and anatoxin. The experimental study is based on bioassays with single and combined N and P additions, including various N species, and 40% dilution of the natural water.
There were variable and inconclusive results in N and P limitation on growth, biomass yield and toxin production. The efficiency of the 40% dilution, mimicking nutrient reduction goals imposed by management authorities, were tested and also gave mixed results. A common feature was that the choice of N species did not affect the results.
As the natural nitrogen levels, and the chlorophyll concentrations in the Western Lake Erie (WLE) are very high, the August experiments may have suffered from extensive bottle effects.
This is one of several recent studies to highlight to role of N availability in shaping cyanobacterial blooms in lakes, shifting the decades long paradigm of exclusive P control.
The introduction is very well written and gives a good account on recent developments, including the effects of reduced nitrogen sources on non-diazotrophic cyanobacterial development and toxin production.
The study has clear research questions and although the experimental outcome is not clear cut, I do not find fundamental flaws in the experimental setup and find the study publishable after some modifications.
Firstly, the authors should attempt to more clearly outline what were the novel findings and take home message for the reader.
E.g. L 232 We found that both major nutrients, P and N, were limiting to CyanoHAB growth and microcystin and anatoxin production in WLE. -- this is just partly correct, as some experiments concluded no-nutrient limitation.
L 233 the 40% reduction of nutrients could slow growth rates and microcystin production during nutrient replete conditions. -- true, yet an example of the inconclusive outcome.
I find the conclusions are weakly formulated currently. If the mixed results are the main conclusion, even this can be more clearly spelled out.
The main finding at the Abstract level: The results demonstrate the importance of recognizing spatiotemporal heterogeneity in nutrient limitation in aquatic systems -- is hardly new nor the main topic of this study.
The departure point of the 40% dilution treatment was the US Environmental Protection Agency (EPA) and Environment and Climate Change Canada recommendation of a 40% reduction in P loading into WLE to help control the blooms [40–42]. -- Any comment from where the 40% came from? I can see that this 40 determined the experiemntal treatments in this study. We commonly refer that policy should be science based. Here the roles are reversed, the research is policy based. This leaves us in uncertainty -- would 35% be as effective (but less costly) or would 50% do a considerably better job.
The topic is a bit touched in the conclusion, but here mostly 40% reduction in the N loading, which is quite different from the EPA recommendation.
Specific comments:
Introduction.
L 60, perhaps reveal the N:P ratio of manure based fertilizers
L 93 "Additionally, anatoxins are also enriched in N [39]." -- redundant, the content is given in the previous sentence already.
L 104 -- research question 2: are nutrient limitation dynamics similar to those controlling CyanoHAB biomass production? -- this can be re-formulated for clarity. Are nutrient availability/limitation effects on toxin and biomass production similar or different.
Research questions 3 and 4 have a lot of overlap.
Results
L 129 in the without biomass normalization -- this is a bit weird construct, particularly, as the Method section comes later on. Perhaps try to re-word? Biomass-normalized analyses is clear.
Figs 2-7 Figures can be improved: drop fractional days in x-axis tick labels,
perhaps also make lines better visible and distinguishable by increasing the size of symbols.
Panel labels (a-f) are not visible (technical issue)
Except for Fig 5, the upper two panels on right column seem quite poor in information. Maybe omit?
Fig. 2 legend on the right side: F = 0.2.513 - is this correct?
Also, seems the dilution anova the non-diluted samples were contrasted against the diluted ones (df = 1). This was not spelled out in the methods. It is a bit unclear how the test was performed, does it correspond to paired t-test?
Fig 5 please check the labels. On panels e and f there are twice Controls and no urea treatment. Panel f the bars do not seem to correspond to the lines in panels b and d (e.g. N+P show decrease in bar graph, but increase in line graphs)
L 175 The growth rate was significantly affected by nutrients (p < 0.001) -- if this refers to Fig 5, then the nutrient effect is hardly important ecologically -- all lines on panel a and c seem very similar.
Tables 1 and 2 - the nutrient concentrations are 2-3 digit precision, but there seems to be large discrepancy between the undiluted and 40% diluted concentrations. Frequently the diluted concentrations are higher (ammonium in Maumee Bay, Table1; all nutrients in Sandusky Bay, Table 2). This needs to be explained.
Discussion
L 233 We also found that the 40% reduction of nutrients could slow growth rates and microcystin production during nutrient replete conditions (Figures 2F, 3E, and 3F).
Fig 2F does not reveal much difference in growth rates between undiluted and 40% diluted treatments
L 253-255 A complicated sentence and could be re-worded for clarity.
Methods
L 344 Because the EPA Action Plan recommends a 40% input reduction in P, we investigated the effects of a 40% reduction in N -- this is confusing. Didn't you investigate 40% reduction in both, N and P, with the dilution treatment?
L 426 I do not think the interpretation of the production rate, as specified in equation 2, is correct. You can check this out. A doubling per day gives log(2/1) = 0.69, not 1.0. To get the doublings per day from the exponential production rate, one has to divide the production rate with ln(2).
Author Response
Thank you for this very helpful review enabling us to improve the manuscript. We really appreciate it.
Point 1: Firstly, the authors should attempt to more clearly outline what were the novel findings and take home message for the reader.
E.g. L 232 We found that both major nutrients, P and N, were limiting to CyanoHAB growth and microcystin and anatoxin production in WLE. -- this is just partly correct, as some experiments concluded no-nutrient limitation.
L 233 the 40% reduction of nutrients could slow growth rates and microcystin production during nutrient replete conditions. -- true, yet an example of the inconclusive outcome.
Response 1: The outcomes dealt inclusive have been reworded to be more conclusive and to highlight the novel findings and the take home messages. Please see edits in lines 238-241.
Point 2: I find the conclusions are weakly formulated currently. If the mixed results are the main conclusion, even this can be more clearly spelled out.
Response 2: We reworded and reformulated the conclusion, which is now L 326-349. This should strengthen the conclusion by highlighting the key take away points and conclusions of the manuscript.
Point 3: The main finding at the Abstract level: The results demonstrate the importance of recognizing spatiotemporal heterogeneity in nutrient limitation in aquatic systems -- is hardly new nor the main topic of this study.
Response 3: The last sentence of the abstract (L15-18) has been changed to focus on nutrient limitation of microcystin, anatoxin, and biomass in WLE as anatoxin is novel in the WLE blooms. This is one, if not the only, experiment using dilution bioassays to investigate the 40% P reduction (in this case paired with 40% N reduction).
Point 4: The departure point of the 40% dilution treatment was the US Environmental Protection Agency (EPA) and Environment and Climate Change Canada recommendation of a 40% reduction in P loading into WLE to help control the blooms [40–42]. -- Any comment from where the 40% came from? I can see that this 40 determined the experiemntal treatments in this study. We commonly refer that policy should be science based. Here the roles are reversed, the research is policy based. This leaves us in uncertainty -- would 35% be as effective (but less costly) or would 50% do a considerably better job.
The topic is a bit touched in the conclusion, but here mostly 40% reduction in the N loading, which is quite different from the EPA recommendation.
Response 4: The 40% P load reduction was the result of a multi-model exercise apart of the revised Great Lakes Water Quality Agreement between the US and Canada. This is now explained at lines 102-104.
We studied eutrophic bays of Lake Erie and found that 40% reduction might not be adequate in these waters, however, we did not investigate the open waters. We highlighted that an adaptive management approach is needed to determine if the 40% reduction is adequate for the eutrophic bays and the open waters. Future research is needed.
We expanded upon future research looking at dilutions other than 40% in the conclusions, particularly L 332-336. We also included a reference to the Maccoux et al. 2016 paper which contains the baseline P data used for the assessment of P reduction in L 101-103.
Point 5: L 60, perhaps reveal the N:P ratio of manure based fertilizers
Response 5: This sentence was reworded for clarity and the N:P ratio range of manure was added L 61-63.
Point 6: L 93 "Additionally, anatoxins are also enriched in N [39]." -- redundant, the content is given in the previous sentence already.
Response 6: This sentence has been removed with the reference being moved to the prior sentence, which is from L 92-95.
Point 7: L 104 -- research question 2: are nutrient limitation dynamics similar to those controlling CyanoHAB biomass production? -- this can be re-formulated for clarity. Are nutrient availability/limitation effects on toxin and biomass production similar or different.
Research questions 3 and 4 have a lot of overlap.
Response 7: Question 2 has been reworded per your suggestions. Thank you for the very useful advice. Question 3 is in reference to the EPA recommendations and Question 4 is extending Question 3 to both N and P. Please see the edits in L 108-110.
Point 8: L 129 in the without biomass normalization -- this is a bit weird construct, particularly, as the Method section comes later on. Perhaps try to re-word? Biomass-normalized analyses is clear.
Response 8: This sentence has been reworded to a better sentence construction, and the changes are on L 136-137.
Point 9: Figs 2-7 Figures can be improved: drop fractional days in x-axis tick labels,
perhaps also make lines better visible and distinguishable by increasing the size of symbols.
Panel labels (a-f) are not visible (technical issue)
Except for Fig 5, the upper two panels on right column seem quite poor in information. Maybe omit?
Response 9: Figures 2-7 have x-axis tick labels with fractional days removed, lines and symbol sizes have been increased, and the a-f have been moved and made larger. The updated code has been pushed as a new update to the GitHub Repository associated with the paper. For figures 2 and 5, the upper two right panels are Sandusky Bay biomass paralleling the Maumee Bay biomass panels. For Figures 3-4 and 6-7, the upper two right panels are displaying biomass normalized toxin production. If need be, those panels can be removed, but the panels provide a visualization of the change in biomass normalized toxin ratios over the course of the experiments as well as a visualization of the error.
Point 10: Fig. 2 legend on the right side: F = 0.2.513 - is this correct?
Also, seems the dilution anova the non-diluted samples were contrasted against the diluted ones (df = 1). This was not spelled out in the methods. It is a bit unclear how the test was performed, does it correspond to paired t-test?
Response 10: This F value was an issue based on how the composite figures were put together. It should read F = 2.513 and has been updated in Figure 2. Additional information about df in the ANOVA analyses has been added in L470-471 as part of the section laying out how the ANOVA analyses were conducted (L467-473).
Point 11: Fig 5 please check the labels. On panels e and f there are twice Controls and no urea treatment. Panel f the bars do not seem to correspond to the lines in panels b and d (e.g. N+P show decrease in bar graph, but increase in line graphs)
Response 11: This was an error in the code for Figure 5 as it was using the June x labels rather than the August x labels as August had the addition of a Urea treatment. This has since been fixed and should correct the issues brought up in this point.
Point 12: L 175 The growth rate was significantly affected by nutrients (p < 0.001) -- if this refers to Fig 5, then the nutrient effect is hardly important ecologically -- all lines on panel a and c seem very similar.
Response 12: We agree with the assessment and have added the phrase “leading to a lack of ecological significance” to L 182-183.
Point 13: Tables 1 and 2 - the nutrient concentrations are 2-3 digit precision, but there seems to be large discrepancy between the undiluted and 40% diluted concentrations. Frequently the diluted concentrations are higher (ammonium in Maumee Bay, Table1; all nutrients in Sandusky Bay, Table 2). This needs to be explained.
Response 13: Tables 1 and 2 were edited to 2 digit precision. Additionally, text was added to the results section in reference to explanations about the higher nutrient values in the 40% diluted nutrients being higher in L 249-252 and L 254-258.
Point 14: L 233 We also found that the 40% reduction of nutrients could slow growth rates and microcystin production during nutrient replete conditions (Figures 2F, 3E, and 3F).
Fig 2F does not reveal much difference in growth rates between undiluted and 40% diluted treatments
Response 14: For these sentences, now L 238-241 there was some rewording to drive home the key points. Additionally, we removed “2F” and “growth rates” from the sentence to make it more accurate.
Point 15: L 253-255 A complicated sentence and could be re-worded for clarity.
Response 15: The sentence, now on L 266-267, now has the hyphenated section removed and a few other words changed to help with clarity.
Point 16: L 344 Because the EPA Action Plan recommends a 40% input reduction in P, we investigated the effects of a 40% reduction in N -- this is confusing. Didn't you investigate 40% reduction in both, N and P, with the dilution treatment?
Response 16: This is a great point that we needed to clarify in the methodology. Thank you for this suggestion. The existing section has been updated to better clarify the 40% dilutions (L 367-371) with the addition of a qualifying sentence in L 369.
Point 17: L 426 I do not think the interpretation of the production rate, as specified in equation 2, is correct. You can check this out. A doubling per day gives log(2/1) = 0.69, not 1.0. To get the doublings per day from the exponential production rate, one has to divide the production rate with ln(2).
Response 17: Thank you for pointing out this error. Please see the text changes in L 452-454 to remedy this issue.